# Digital Exhibition of Intangible Heritage and the Role of Museums in COVID-19 Era—Focusing on Gwangju Chilseok Gossaum Nori Video Experience Center in South Korea

**Mira Han [1] and Yumi Yi [2,*]**

[1]  Da Vinci College of General Education, ChungAng University, 84 Heuksuk-ro, Seoul 06974, Korea
[2]  Humanities Research Institute, ChungAng University, 84 Heuksuk-ro, Seoul 06974, Korea
*   Correspondence: joystu@cau.ac.kr; Tel.: +82-02-820-5715

**Abstract:** A major purpose of this study is to evaluate the effectiveness of digital exhibitions of intangible heritage in the COVID-19 era and to ascertain the role of museums in enhancing understanding and interest in intangible heritage. This survey was designed to determine Gwangju citizens' perception of intangible heritage and preference for visiting museums, focusing on the Gwangju Chilseok Gossaum Nori Experience Center in Korea. As a result, three perspectives were derived. First, exhibitions and video experience centers that use digital technology were more helpful toward understanding Gwangju Chilseok Gossaum Nori than festivals. Second, the frequency of visits to local festivals or museums positively affected the frequency of visits to the Gwangju Chilseok Gossaum Nori Video Experience Center. This demonstrates that cultural heritage conservation programs have a positive effect on each other. Third, the purpose of visiting the museum varied by age. For example, parents visited museums for educational purposes, which clearly shows that the reason for visiting the museum varies according to age. These results prove that museums must not only understand visitors' needs, but they also must jointly consider ways to revitalize their exhibitions.

**Keywords:** Gwangju Chilseok Gossaum Nori; video experience center; festival; digital technology; Gwangju citizens; level of awareness; level of participation

## 1. Introduction: Exhibition of Intangible Heritage Using Digital Technology

This study is to evaluate the effectiveness of digital exhibitions of intangible heritage in the COVID-19 era and to find the role of museums to enhance understanding and interest in intangible heritage. This survey was designed to find out the perception of the intangible heritage of Gwangju citizens and their preference for visiting museums, focusing on the Gwangju Chilseok Gossaum Nori Experience Center in South Korea.

The conservation of intangible cultural heritage entails passing down its cultural significance and features to the subsequent generation. In this regard, it is critical to secure the possibility of spreading intangible cultural heritage by providing the general public with opportunities to experience and participate in it, as well as to foster bearers and experts. In other words, for the conservation of intangible heritage, informing the general public about its meaning and providing them with opportunities to experience it is crucial [1].

In this respect, the government and intangible cultural heritage protection organizations are using various methods, such as festivals and exhibitions, to enable opportunities for the general public to experience—and learn about—their intangible heritage. Moreover, recently, the number of museums where intangible heritage can be accessed at all times is increasing. Unlike tangible heritage, intangible heritage—such as music, dance, and craftsmanship—possesses neither form nor shape. Considering these features, the use of digital technologies, such as virtual reality (VR) and augmented reality (AR), has been emphasized in recently established museums displaying intangible heritage [2,3].

Museums that use technologies, such as VR and AR, to exhibit intangible heritage in South Korea are the Video Experience Center, exhibiting Gwangju Chilseok Gossaum Nori, the traditional Korean loop fight; and the National Intangible Heritage Center, exhibiting Jultagi, tightrope walking, and Tal Nori, mask plays. As the exhibition of intangible heritage through digital technology is presented as a sustainable way to conserve heritage, it is expected that this type of exhibition will gradually increase in the future. However, it is doubtful whether the varied intangible heritage contents created by digital technology will change the public's perception about them. While existing studies have primarily focused on the use of digital technology on intangible heritage [3], research analyzing the perceptions of the general public, the primary users of digitally created content, and the people visiting museums—where various contents are displayed and experienced [4]—has been insufficient.

Therefore, among the various examples of intangible cultural heritage in South Korea, this study examines ways to promote the Gwangju Chilseok Gossaum Nori Video Experience Center, which is making efforts to conserve and display the subject using digital technologies such as 4D and VR. For the conservation of Gwangju Chilseok Gossaum Nori, Gwangju Metropolitan City hosts a festival and operates the Video Experience Center for on-going exhibitions. However, since the festival is held outdoors for a limited time, there are several impediments for the public's continuous experience of this tradition [5]. Meanwhile, the Video Experience Center is emerging as a solution to this problem. Currently, the Center provides the general public with the opportunity to experience intangible heritage at all times through exhibitions and experiences using digital technologies, such as 4D, VR, and also 2D.

In particular, under conditions where it is difficult to hold festivals, such as a pandemic, the Video Experience Center is widely recognized as an essential means to increase citizens' understanding of intangible heritage. Therefore, the search for ways to raise public awareness of the Video Experience Center using digital technology—the focus of this study—will serve as fundamental data for exploring ways to promote the conservation of intangible heritage using digital technology in the era after the coronavirus disease 2019 (COVID-19) pandemic.

## 2. Exhibition of Gwangju Chilseok Gossaum Nori and the Current Status of the Use of the Video Experience Center

Gossaum Nori is a folk game, which was played extensively at the beginning of the New Year: that is, from the day of the first full moon of the year. It is a game in which a "Go" is built using straws as the primary material; the players are divided into two teams who then compete to win [6]. In the past, it was played in various regions of South Korea; unfortunately, the modern industrialization period interrupted the transmission of the game, which now remains only in the older generation's memories [7]. Hence, the Gossaum Nori, played in Chilseok-dong, Gwangju, was designated as Important Intangible Cultural Property No. 33 in 1970, and Gwangju Chilseok Gossaum Nori (Figure 1) became a group folk game representing South Korea [8].

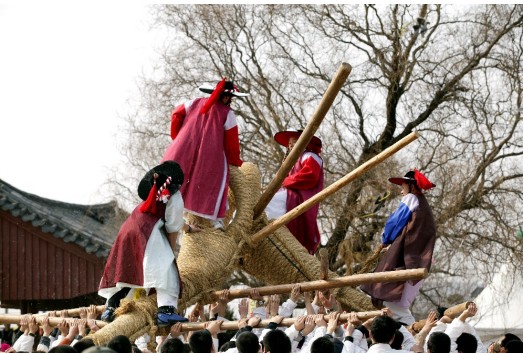

**Figure 1.** Gwangju Chilseok Gossaum Nori (Loop Fight of Gwangju) [9].

However, the process of preserving Gwangju Chilseok Gossaum Nori has not been easy. The years 1983 and 1998 witnessed the death of the master performers of the game. Further, the Gossaum Nori community, which originally comprised 185–208 members for each team, has been reduced due to a decrease in the population of the local society. Additionally, the rapid spread of COVID-19 in 2020 brought to the forefront other challenges regarding opening the exhibition center and conducting apprenticeship training. Despite these difficulties, efforts are being made to conserve Gwangju Chilseok Gossaum Nori; these efforts include various changes in: the exhibition methods, hosting of Gossaum Nori festivals, and continuous training for apprenticeship in response to social changes [10].

In Gwangju, the "Gosssaum Nori Festival" has been held annually since 1982. The festival is meaningful because it promotes this intangible heritage and provides people with a firsthand experience [11]. However, since it is not a year-round performance, there were not enough opportunities for the public to experience it. Hence, the Gossaum Nori Theme Park, which allows the public to experience Gosssaum Nori throughout the year, was opened in Chilseok-dong, Nam-gu, Gwangju in 2009. The Gossaum Nori Theme Park, currently, comprises a Video Experience Center and a Training Center for apprenticeship. The former opened in February 2010 and was planned to be a place for a year-round virtual experience of Gossaum Nori, which is only performed on the day of the first full moon of the year, through advanced media. It aims to transmit not only the game's meaning but also the tradition [12,13]. The Video Experience Center can largely be divided into an exhibition zone and an experience zone. The exhibition zone displays information related to the historical origins of Gossaum Nori, the process of producing the Go, and the participants in it. The experience zone includes 4D video experiences of Gossaum Nori, and is a place for in-depth learning of this traditional game through a computer. In 2018, an experience involving the application of VR technology to Gwangju Chilseok Gossaum Nori was added. Until recently, intangible heritage has been difficult for the public to access due to the challenges of presenting an exhibition or facilitating successor-centered activities [14]. The Gwangju Chilseok Gossaum Nori Video Experience Center, which contains a variety of contents, has been meaningful in the sense that it allows the public to understand and experience the game. This center is particularly significant because it creates content by applying new technologies to this folk game, and provides it to the general public.

However, currently, the Video Experience Center has not been used to its fullest extent. In 2018 and 2019, 5641 and 6515 people, respectively, used the 3D theater, and 2500 and 2770 people, respectively, used the VR experience room. These user numbers prove that both the 3D Video Center and VR Experience Center are not being actively operated. Therefore, this study examines ways to promote the Video Experience Center through a survey of Gwangju citizens' perceptions of Gwangju Chilseok Gossaum Nori.

The Gwangju Chilseok Gosaum Nori video experience center is meaningful in that it is a space where citizens can experience intangible heritage.Methods such as festivals to preserve intangible heritage became impossible due to COVID-19, and new attempts were needed. COVID-19 has caused various problems, from conservation and management of intangible heritage to citizens' tourism [15]. The Gangju Chilseok Gosaum Nori Video Experience Center complements the problem of preservation methods such as festivals through the method of the museum. However, COVID-19 has also become a problem for museums. When the museum was closed due to the pandemic, the museum began to try new ways of operating it. The Korean museum showed a new change in the management of the museum by attempting to communicate with citizens using digital in a situation where the museum was closed due to the pandemic [16]. The important point in the new museum's attempt is that after COVID-19, museums began to actively use online programs [17,18]. The online use of the museum not only expanded the exhibition space, but also applied digital exhibitions. VR and AR were used to improve the visitor experience [19]. In addition, through research on the preference for content that needs digitalization [20], exhibitions and experiences using digital content were attempted in museums. Gwangju Chilseok Gosaum Nori Video Experience Center also planned various

digital experiences using VR or 4D movie theaters as well as experiences such as external theme parks and festivals. This plan made it possible to experience Gwangju Chilseok Gosaum Nori, although limited, by using a video experience center during the pandemic.

### 3. Relationship between Level of Awareness and Perception of Exhibition Methods of Gwangju Chilseok Gossaum Nori

A questionnaire survey was administered to confirm Gwangju citizens' understanding of Gwangju Chilseok Gossaum Nori and their participation in it, including their perception of the effectiveness of various exhibition methods. The survey was administered for about two weeks from 19 October to 9 November 2020. Survey participants included 196 Gwangju citizens (living in Gwangju for more than one year), aged 20–50 years. (The survey respondent information is shown in Table 1). The survey was limited to Gwangju citizens because the festivals, exhibiting Gwangju Chilseok Gossaum Nori; the Video Experience Center, providing exhibitions and experiences using digital technology; and the Education Center, for cultural transmission; are all conducted or located in Gwangju. Thus, the accessibility of Gwangju citizens is assumedly higher than that of people from other regions. In other words, despite the geographical advantages, since the accessibility and utilization rate of people from other regions must be lower than that of local residents, we decided to evaluate Gwangju citizens' utilization rate and awareness as the first step to confirm the public's perception of Gwangju Chilseok Gossaum Nori[1].

**Table 1.** Survey Respondents' Information.

| Gender | N | Parental | N | Marital Status | N | Age | N | Residence Period in Gwangju | N | Age of Children | N |
|---|---|---|---|---|---|---|---|---|---|---|---|
| Male | 114 | Children | 99 | Married | 96 | 20s | 39 | Less than 1 year | 1 | Under 7 years | 18 |
| Female | 81 | No children | 97 | Single | 88 | 30s | 49 | Over 1 year-Less than 5 years | 10 | 8–13 years | 18 |
| No response | 1 | | | Other | 12 | 40s | 50 | Over 5 year-Less than 10 years | 14 | 14–16 years | 10 |
| | | | | | | 50s | 58 | Over 10 year | 169 | 17–19 years | 9 |
| | | | | | | | | No response | 2 | Over 20 years | 44 |
| Total | 196 | | 196 | | 196 | | 196 | | 196 | | 99 |

The survey comprised 12 questions to confirm Gwangju citizens' levels of awareness and utilization rates regarding Gwangju Chilseok Gossaum Nori. First, through 6 questions, we tried to confirm their understanding of Gwangju Chilseok Gossaum Nori; their perception of various public-oriented exhibition methods, such as the festival and the Video Experience Center; and their frequency and intention of visiting them. These specific questions are shown in Table 2. Additionally, participants were questioned about the path through which they became acquainted with the folk game, measures to promote public awareness, participation in the Video Experience Center and the festival of Gwangju Chilseok Gossaum Nori, its exhibition methods, and their reasons for visiting the museum. In this study, citizens' participation in festivals was investigated to study digital exhibition methods because festivals are the most common methodology in preserving intangible heritage. Seo H.J.'s research shows that festivals are an important conservation method. As a result of network analysis of the latest issues in the field of intangible cultural heritage using text mining, only 'festival' was included among the top 52 keywords [21].

Gwangju citizens' levels of awareness of Gwangju Chilseok Gossaum Nori, participation in the festival, and use of the Video Experience Center were analyzed using SPSS 25.0. The Cronbach's alpha value[2] for 12 questions, excluding survey respondent information, was 0.674, and that for the above-mentioned 6 questions was 0.842, thus showing a very high reliability.

**Table 2.** Survey Questions about Gwangju Citizens' Levels of Awareness and Rates of Utilization Regarding Gwangju Chilseok Gossaum Nori.

| |
|---|
| Q1. Do you know about Gwangju Chilseok Gossaum Nori, which is an intangible cultural heritage? |
| Q2. Do you think Gwangju Chilseok Gossaum Nori Festival will help Koreans understand Gwangju Chilseok Gossaum Nori? |
| Q3. Do you think Gwangju Chilseok Gossaum Video Experience Center (4D movie and VR experience) will help Koreans understand Gwangju Chilseok Gossaum Nori? |
| Q4. Do you often play online games (2D, 3D, VR, etc.)? |
| Q5. Do you visit local museums often? |
| Q6. Do you attend local festivals often? |

*3.1. Gwangju Citizens' Level of Awareness Regarding Gwangju Chilseok Gossaum Nori*

The level of awareness score of Gwangju Chilseok Gossaum Nori was 2.73 out of 5[3]. Despite the fact that this survey was only administered to Gwangju citizens, a high score was not obtained. This confirms the need to undertake additional efforts to raise awareness about the traditional game. To examine Gwangju citizens' awareness level in more detail, we analyzed whether there is a difference based on sociodemographic characteristics. Independent samples *t*-test[4] and ANOVA[5] were used to confirm the variables—gender, marriage, children, and age—to determine how the differences between the means appeared.

Table 3 presents differences in the average awareness level of Gwangju citizens. Males, married individuals, and people with children showed higher levels of awareness regarding Gwangju Chilseok Gossaum Nori compared to females, single individuals, and people without children, respectively. Further, the level of awareness increases with age. Gwangju Chilseok Gossaum Nori, an intangible cultural heritage, is part of the elementary school curriculum in South Korea[6]. However, the results confirm the limitations of raising awareness through the curriculum. Furthermore, this survey confirmed the difference in awareness of the folk game between participants and non-participants of festivals or the Video Experience Center. Thus, an individual's experience and social conditions exert a significant influence on the formation of cultural heritage awareness. In addition to the questions presented in Table 2, our survey included the following questions: "Have you ever participated in Gwangju Chilseok Gossaum Nori Festival?" and "Have you ever visited Gwangju Chilseok Gossaum Nori Video Experience Center?" Only 11.2% and 11.7%, respectively, had experienced the festival and the Video Experience Center. However, examining the difference in the level of awareness—between those who have experienced the festival or the Video Experience Center and those who have not—revealed that the former group showed a higher average awareness level (4.05 and 3.87 for the festival and the Video Experience Center, respectively). This confirms that exhibitions, such as the festival and the Video Experience Center, affect citizens' awareness of Gwangju Chilseok Gossaum Nori.

**Table 3.** Awareness of Gwangju Chilseok Gossaum Nori.

| | | | Q1 |
|---|---|---|---|
| | | Mean $\pm$ SD | *p* |
| Gender | Male | 2.90 $\pm$ 1.024 | 0.022 * |
| | Female | 2.48 $\pm$ 1.333 | |
| Marriage | Married | 3.06 $\pm$ 1.186 | 0.000 * |
| | Single | 2.36 $\pm$ 1.270 | |
| Children | Children | 3.11 $\pm$ 1.192 | 0.000 * |
| | No Children | 2.35 $\pm$ 1.236 | |
| Age | 20s | 1.95 $\pm$ 1.169 | |
| | 30s | 2.55 $\pm$ 1.339 | 0.000 * |
| | 40s | 3.10 $\pm$ 1.074 | |
| | 50s | 3.09 $\pm$ 1.189 | |

* $p < 0.05$.

### 3.2. Perceptions on the Exhibition Methods for Gwangju Chilseok Gossaum Nori

To assess whether the various exhibition methods of Gwangju Chilseok Gossaum Nori affect its understanding, the survey asked questions (Q2, Q3) focusing on the festival and the Video Experience Center. Unlike a simple awareness-level survey on Gwangju Chilseok Gossaum Nori, these questions pertained to the methods of exhibiting intangible cultural heritage and the effectiveness of the festival and Video Experience Center in understanding such heritage. In response to these items, the average scores for the festival and the Video Experience Center were 3.67 and 3.76, respectively. Such high scores indicate that both the festival and Video Experience Center enable the understanding of Gwangju Chilseok Gossaum Nori.Between the festival and Video Experience Center, the latter, which provides various exhibitions and experiences using digital technology, obtained a higher average value. Notably, the Video Experience Center is more helpful in understanding Gwangju Chilseok Gossaum Nori. In particular, As you can see in Table 4, both the festival and the Video Experience Center were helpful in understanding Gwangju Chilseok Gossaum Nori. However, while the festival seemed to affect the level of understanding equally regardless of sociodemographic variables, the Video Experience Center proved more helpful for married people and people with children. According to previous research in the United States, about 80% of museums provide educational programs for children [22] and spend more than $2 billion a year on education activities. therefore parents who participated in the survey also expect the visit to the museum to affect their children's education [23].

**Table 4.** Differences between Means Depending on Sociodemographic Characteristics.

| | | Q2 | | Q3 | |
|---|---|---|---|---|---|
| | | **Mean ± SD** | ***p*** | **Mean ± SD** | ***p*** |
| Gender | Male | 3.76 ± 0.834 | 0.092 | 2.90 ± 1.024 | 0.081 |
| | Female | 3.55 ± 0.925 | | 2.48 ± 1.333 | |
| Marriage | Married | 3.77 ± 0.888 | 0.227 | 3.06 ± 1.186 | 0.025 * |
| | Single | 3.62 ± 0.775 | | 2.36 ± 1.270 | |
| Children | Children | 3.79 ± 0.888 | 0.066 | 3.11 ± 1.192 | 0.003 * |
| | No Children | 3.55 ± 0.850 | | 2.35 ± 1.236 | |
| Age | 20s | 3.57 ± 0.698 | | 1.95 ± 1.169 | |
| | 30s | 3.55 ± 1.081 | 0.301 | 2.55 ± 1.339 | 0.055 |
| | 40s | 3.66 ± 0.895 | | 3.10 ± 1.074 | |
| | 50s | 3.84 ± 0.745 | | 3.09 ± 1.189 | |

* $p < 0.05$.

This result can be interpreted as a consequence of the Video Experience Center possessing flexible accessibility and the educational function of a museum. Festivals are limited in that they are only held during a specific period. Moreover, accurately understanding the subject in a festival is difficult, and parents carry the burden of providing information to their children[7]. The Video Experience Center was more helpful in understanding Gwangju Chilseok Gossaum Nori because it provides detailed information regarding this intangible heritage. However, since the characteristics of these exhibition methods serve different purposes, it is not possible to claim one's superiority over the other. The data merely explain which exhibition method is easier for the general public to understand and access the target subject.

### 3.3. Relationship between Awareness and Positive Perception of the Exhibition Methods of Gwangju Chilseok Gossaum Nori

Correlation analysis[8] was conducted to confirm the relationship between knowing about Gwangju Chilseok Gossaum Nori and the perception of exhibition methods—the festival and Video Experience Center—being helpful in understanding Gwangju Chilseok Gossaum Nori. Table 5 shows the results of correlation analysis.

**Table 5.** Correlation between Questions.

|      | Q1 | Q2        | Q3        | Q4      |
|------|----|-----------|-----------|---------|
| Q1   | 1  | 0.310 **  | 0.301 **  | 0.072   |
| Q2   |    | 1         | 0.761 **  | 0.133   |
| Q3   |    |           | 1         | 0.149 * |
| Q4   |    |           |           | 1       |

\* $p < 0.05$ ** $p < 0.01$.

Table 5 shows that most of the questions on the level of awareness and understanding of Gwangju Chilseok Gossaum Nori are correlated. Looking at the correlation[9] results, positive perceptions of the festival (Q2) and the operation of the Video Experience Center (Q3) showed a strong correlation with improvements in the understanding of the game ($r = 0.761$, $p = 0.000$). As shown in Tables 3 and 4, the average awareness was on a moderate level at 2.73. In addition, of the 196 survey respondents, only 22 (11.2%) had experience participating in the festival, and 23 (11.7%) had experience using the Video Experience Center. However, the average awareness was 4.05 for those who had participated in the festival, higher than 2.56 for non-participants ($p = 0.000$) and 3.81 for visitors to the Video Experience Center, which was higher than that of non-visitors at 2.58 ($p = 0.000$). These results show that direct experiences such as participation in the festival and Video Experience Center acted as an important factor in the awareness of cultural heritage. This is meaningful in that the experience of visiting the museum using VR and AR not only led to the museum's revisit, but also confirmed that external events related to the museum's exhibition theme were related to the museum's revisit [24]. In this context, encouraging the general public to participate in the festival and providing more opportunities to experience Gwangju Chilseok Gossaum Nori through the Video Experience Center can be a key method to promote their awareness of this traditional game.

## 4. Revitalization of Gwangju Chilseok Gossaum Nori Video Experience Center

### 4.1. Factors Influencing Visits to Gwangju Chilseok Gossaum Nori Video Experience Center

In terms of exhibiting Gwangju Chilseok Gossaum Nori, the festival and Video Experience Center have their own unique advantages and disadvantages. On the one hand, the festival only takes place during a specific period; thus, the amount of time for public access is limited. Another disadvantage is that the event may not be held at all due to the absence of apprentices or circumstantial reasons such as COVID-19. However, since the festival is held in a wide-open space, it is more accessible to a large number of people than the Video Experience Center. Therefore, it has a positive effect on the awareness of citizens. On the other hand, the Video Experience Center has the disadvantage that it is relatively difficult for the public to access without active publicity because exhibitions and experiences are held in a limited space. However, it is meaningful in that it can be accessed year-round by visitors, who can enjoy a firsthand experience of various Gwangju Chilseok Gossaum Nori contents produced using digital technology [25–27].

Our survey confirmed that Gwangju citizens' level of awareness of the folk game is low. At the same time, the benefits of exhibitions, such as the festival and Video Experience Center, have not been fully realized. However, considering that as the utilization of the festival or Video Experience Center increased, the understanding and awareness of Gwangju Chilseok Gossaum Nori increased parallelly, frequently exposing the general public to exhibitions, such as the festival and Video Experience Center, is critical to increasing their understanding and awareness. Unfortunately, during crises, such as the COVID-19 pandemic, it is difficult to host external events, such as festivals, without interruptions. In addition, since festivals are held only for a limited time and are significantly affected by the surrounding circumstances, video experience centers, which constitute an exhibition method that uses digital technology, are gaining increased attention as an alternative. Nonetheless, due to a lack of understanding of intangible heritage and limited space, the general public's accessibility to the Video Experience Center is insufficient. Therefore, by

analyzing the factors that influence the intention of visiting the Video Experience Center, we attempt to find a way to increase its accessibility to the public.

To analyze the factors influencing the intention to visit the Gwangju Chilseok Gossaum Nori Video Experience Center, the model fit was confirmed in a stepwise selection method[10] to compute multiple regression analysis. Durbin-Watson, tolerance limits, and variance inflation factor (VIF) values were calculated to determine whether the regression analysis outputs met the basic assumptions. The Durbin-Watson value to test the independence of the residual was 1.700, which was within the normal range (1.5–2.5); furthermore, the tolerance limit for evaluating the multiple collinearity was 0.740–0.848, and the VIF was 1.351–1.179, which did not deviate from the normal range of less than 1 and between 1 and 9, respectively, thus satisfying the basic assumptions for regression analysis. The coefficient of determination of the model was 51.2% (=0.512).

As shown in Table 6, Model 3—including Q3, Q5, and Q6—was adopted. Factors influencing the intention to visit the Gwangju Chilseok Gossaum Nori Video Experience Center were as follows: the degree to which the Video Experience Center helped understand Gwangju Chilseok Gossaum Nori (Q3), frequency of visits to museums (Q5), and frequency of participation in local festivals (Q6). According to regression model 3, if the degree to which the Video Experience Center helped understand Gwangju Chilseok Gossaum Nori increased by 1 point, intention to visit the Center increased by 0.541 points on average. Further, if the frequency of visits to museums increased by 1 point, the intention to visit the Video Experience Center increased by 0.256 points on average; thus, it can be predicted that if the frequency of regular participation in local festivals increases by 1 point, the intention to visit the Video Experience Center will increase by 0.207 points on average.

**Table 6.** Analyzing the Factors Affecting Visits to the Gwangju Chilseok Gossaum Nori Training and Experience Centers.

| Variables | Model 1 | | | | Model 2 | | | | Model 3 | | | |
|---|---|---|---|---|---|---|---|---|---|---|---|---|
| | B | SE | β | p | B | SE | β | p | B | SE | β | p |
| Q3 | 0.700 | 0.073 | 0.638 | 0.000 *** | 0.575 | 0.072 | 0.525 | 0.000 *** | 0.541 | 0.072 | 0.494 | 0.000 *** |
| Q5 | | | | | 0.337 | 0.069 | 0.324 | 0.000 *** | 0.256 | 0.074 | 0.246 | 0.001 *** |
| Q6 | | | | | | | | | 0.207 | 0.080 | 0.182 | 0.011 |
| $F(p)$ | 90.596 (0.000) | | | | 65.199 (0.000) | | | | 47.604 (0.000) | | | |
| $R^2$ | 0.403 | | | | 0.491 | | | | 0.512 | | | |

*** $p < 0.001$.

This result confirms that exerting efforts, such as promotions and events, to increase Gwangju citizens' participation in various museums and festivals in the Gwangju area, positively influenced their intention to visit the Video Experience Center and increased their levels of awareness. Therefore, a greater positive impact can be achieved by organizing promotions and events in conjunction with various museums and festivals in the Gwangju area rather than arranging independent promotions or events to encourage the general public to visit the Video Experience Center. Thus, efforts toward increasing citizens' accessibility to various museums and festivals in the Gwangju area are as important for Gwangju Chilseok Gossaum Nori as for other cultural heritage of the Gwangju area in general.

*4.2. Characteristics of Museum Visitors*

Most museum research related to citizens focuses on educational programs such as examining the value of educational programs and effectiveness of methodologies [28] When studying revitalization plans through cooperation with museums, the linkage to museum materials is generally focused on [29]. However, since the revitalization of museums is closely connected to the utilization rate of visitors, it is important to identify visitors' characteristics. Understanding visitors' characteristics—such as whether they visit museums to learn about a specific topic or simply because of their interest in a place

called a museum—will have an important influence on museum promotion and program planning [30,31].

The Video Experience Center, a method used to exhibit Gwangju Chilseok Gossaum Nori using digital technology, has the characteristics of a museum. However, of the 196 survey respondents, only 23 (11.7%) visited the Center, which confirms that the current awareness about—and visits to—the Video Experience Center are very low. In this regard, Table 6 presents factors influencing the intention to visit the Video Experience Center—examined through regression analysis. We found that a higher frequency of visiting museums led to a higher intention to visit the Video Experience Center. Therefore, we attempted to analyze the purposes and intentions of people visiting museums. The Video Experience Center shares certain characteristics with museums (e.g., exhibitions and experiences of intangible cultural heritage). Hence, confirming the intention of the general public to visit museums would be helpful not only to suggest a direction to increase visits to the Video Experience Center, but also to create a "virtuous cycle" effect in terms of increasing the frequency of visiting the Center by increasing the frequency of visits to other museums.

Therefore, the personal purposes and reasons for visiting museums were investigated through a questionnaire, and the relevance of the age group was checked through the chi square test[11] ($x^2$ test; $p = 0.001$).

Table 7 indicates that there were clear differences in reasons for visiting museums by age group. Visits to acquire historical knowledge were highest in the people in their 50s, while visits for their children's education were highest in the people in their 30s and 40s. The purpose of experiencing a variety of activities was high among those in their 20s and 50s [32]. These results show that different age groups had different reasons for visiting the museum. This result is also related to Powell & Kokkranikal (2015)'s claim that Museum provid an opportunity for a good day out has more appeal to the visitors than the collections in the museum for the average visitors [33]. Thus, the necessity of developing various museum programs to suit each age group was confirmed. Despite the differences in the reasons for visiting museums by age, the item that scored a high percentage among all age groups was "To enjoy leisure time". This confirms citizens' desire to make the museum a space where they can find additional meaning while enjoying their leisure time.

**Table 7.** Reasons for Visiting Museums by Age Groups.

| Age Groups | | To Study History for Him/Herself | To Enjoy Leisure Time | To Educate His/Her Children | To Enjoy Various Experience Activities | Others | Total |
|---|---|---|---|---|---|---|---|
| 20s | N | 4 | 12 | 1 | 12 | 4 | 33 |
| | % | 17.4% | 16.0% | 3.2% | 32.4% | 80.0% | 19.3% |
| 30s | N | 4 | 20 | 12 | 6 | 1 | 43 |
| | % | 17.4% | 26.7% | 38.7% | 16.2% | 20.0% | 25.1% |
| 40s | N | 4 | 18 | 13 | 7 | 0 | 42 |
| | % | 17.4% | 24.0% | 41.9% | 18.9% | 0.0% | 24.6% |
| 50s | N | 11 | 25 | 5 | 12 | 0 | 53 |
| | % | 47.8% | 33.3% | 16.1% | 32.4% | 0.0% | 31.0% |
| Total | N | 23 | 75 | 31 | 37 | 5 | 171 |
| | % | 100.0% | 100.0% | 100.0% | 100.0% | 100.0% | 100.0% |

Currently, a space that exhibits the historical origins of Gossaum Nori using technology, such as graphic panels, is the Gossaum Nori Video Experience Center; however, its composition is focused more on the hall where various hands-on activities, such as 4D and VR experiences, and games using Gossaum Nori, can be performed. The configuration of this video experience center allows visitors to have fun while learning educational content, which, despite its benefits, does not motivate them to come to the museum. To encourage citizens who are interested in history or local folklore to visit the Video Experience Center,

drawing their attention to the various educational programs offered will be important [34]. For those in their 50s, whose purpose of visiting the museum is studying history, the possibility of visiting the Video Experience Center will increase if a history education program is provided. Through this chain reaction, an opportunity to watch the Gwangju Chilseok Gossaum Nori exhibited in the Video Experience Center may be provided[12]. As described above, the development of various programs that meet the needs of the generation can contribute to raising awareness of the Gwangju Chilseok Gossaum Nori as well as other intangible cultural heritage of the Gwangju area.

*4.3. Revitalization Plan for the Exhibition Method of Gwangju Chilseok Gossaum Nori Utilizing Digital Technology*

For a long time, the primary goals of museums were to collect, exhibit, preserve, and study cultural heritage [35] (p. 745). However, museums are beginning to recognize their social roles and responsibilities, such as education and the discussion of social agendas, as well as research on the collection, exhibition, and preservation of cultural heritage [36]. This perspective forces us to think about the importance of marketing methods for visitors, beyond the concerns about museum exhibitions. Gilmore and Rentschler [35] (p. 749) suggested the following three types of services museums provide: education, accessibility, and communication. This is not only a discussion related to the various educational programs currently provided by museums, but also a way to increase museums' accessibility and communication services for various visitors, including local residents and tourists.

It is necessary to think about the revitalization plan of the Video Experience Center in terms of the role of the museum described above. To fulfill its social role, the Video Experience Center, like museums, should have a positive impact on the preservation of cultural heritage by raising awareness of Gwangju Chilseok Gossaum Nori among Koreans, or at least among Gwangju citizens. For this purpose, the use of museums by Gwangju citizens was analyzed according to sociodemographic characteristics. Based on the analysis, the following measures to promote the Video Experience Center may be considered.

First, the Gwangju Chilseok Gossaum Nori Video Experience Center needs educational programs tailored to various age groups [37]. Currently, the program of the Gossaum Nori Video Experience Center is an experience-oriented exhibition due to the special nature of the exhibition using digital technology. However, based on the survey results, each age group has different reasons for visiting; thus, a diversified educational program tailored to various age groups is needed in addition to the activities provided. As shown in Table 7, people in their 20s chose experiential activities as the primary reason for visiting museums, while other age groups chose to visit museums for different reasons[13]. In that sense, the exhibition of cultural heritage using digital technology is much more useful than conventional methodologies. However, the most important aspect of an exhibition is to help citizens remember the subject for a long time. To this end, it is necessary for citizens to visit numerous museums exhibiting intangible cultural heritage. To achieve this goal, programs are needed to motivate citizens to visit museums. Operating a large number of experience programs only for a specific social class without considering the various generations cannot attract a variety of visitors. In other words, varied educational programs, experiences, and leisure spaces that can be enjoyed by each generation should be prepared to make it a space for people of different age groups to visit.

Second, in addition to learning and understanding Gwangju Chilseok Gossaum Nori, a space and program to spend leisure time must be provided. One of the main reported reasons people in their 20s to 50s for visiting the museum was "To enjoy leisure time". Thus, organizing the Video Experience Center as a space where one can enjoy varied leisure activities, which are not restricted to educational purposes, can act as a factor stimulating visits. Currently, the Video Experience Center is a facility within the Gossaum Nori Theme Park, which also includes a folk-game experience park and an outdoor performance stage. However, these spaces are only used for specific events and are not actively utilized in general. Therefore, developing and operating various programs other than those related to

Gossaum Nori, which can be provided in the existing spaces for citizens to enjoy varied leisure activities, will create more opportunities for more citizens to visit the Center.

This view is in line with Doering's [38] argument that defining the role of a museum as a lifelong educator is insufficient to explain the holistic value of the museum experience, and that the role of the museum should be considered from the visitor's point of view by considering the museum-goers as clients. From this perspective, museums need to function as places where visitors can enjoy leisure time more often and comfortably. Additionally, more efforts should be made to provide programs that facilitate access to museums to increase local residents' understanding of cultural heritage.

Third, we must find more ways to increase the overall use of the museum as well as the program for the Video Experience Center by Gwangju citizens.

Table 8 confirms that the frequency of visits to museums and participation in festivals has a strong relationship with the degree to which the exhibition method of Gwangju Chilseok Gossaum Nori is helpful in understanding the folk game (correlation between Q5, Q6 and Q2, Q3). Further, the frequency of visits to museums or participation in festivals showed a strong correlation with the level of awareness of Gwangju Chilseok Gossaum Nori (correlation between Q5, Q6 and Q1). This shows that considering ways to increase the utilization of the entire local museum positively affects not only the utilization of the Video Experience Center, but also the understanding and awareness of Gwangju Chilseok Gossaum Nori. Even if museums are actually designed to display different heritages, if citizens' overall interest in cultural heritage increases, interest in—and the use of—the Gwangju Chilseok Gossaum Nori Video Experience Center will also simultaneously increase. Therefore, it is necessary to work together to devise measures to increase the awareness and utilization rate of Gwangju citizens not only for the Video Experience Center but also for the entire museum.

**Table 8.** Correlation between questions.

| | Q5 | Q6 |
|---|---|---|
| Q1 | 0.335 ** | 0.355 ** |
| Q2 | 0.307 ** | 0.339 ** |
| Q3 | 0.242 ** | 0.268 ** |
| Q4 | 0.243 ** | 0.170 * |
| Q5 | 1 | 0.507 ** |
| Q6 | | 1 |

$* p < 0.05$, $** p < 0.01$.

## 5. Discussion

The purpose of this study is to determine both the effectiveness of digital exhibitions and the role of museums in overcoming the limitations of conventional methods of exhibiting intangible heritage during COVID-19. In particular, the development of digital technology not only reproduces intangible heritage but also helps citizens experience it firsthand. In addition, we investigated why citizens visit the museum. This is because I thought that the reason why citizens visit museums is not only because they are interested in each theme of cultural heritage, but also because they like the museum itself. To confirm this hypothesis, a survey was conducted with the contents of Gangju Chilseok Gosaum Nori and its video experience center. The results are as follows.

1. Digital exhibitions and Video Experience Centers were more helpful than festivals to understand Gangju Chilseok Gosaum Nori.

2. Frequency of visits to local festivals and museums had a positive effect on the frequency of visits to the Gwangju Chilseok Gossaum Nori Video Experience Center, indicating that cultural heritage conservation programs have a positive effect on each other.

3. The purpose of citizens' visit to the museum varied by age. Just as the group with children showed a high understanding of intangible heritage, the main purpose of parents'

visit to the museum was education. Respondents in their 50s, on the other hand, visited for leisure, to enjoy activities, and to study history, while visiting for the purpose of child education was not prevalent.

First, The results that digital exhibitions and video experience centers helped to understand Gangju Chilseok Gosaum Nori rather than festivals are also related to research that Ott, & Pozzi [39] suggested the role of ICT (Intangible Cultural Heritage) in cultural heritage education. Ott, & Pozzi presents Personalized, Inquiry-based learning approaches, Enriched situated learning approaches, and Interdisciplined learning approaches [39] as ICT roles in cultural heritage education. This means that cultural heritage education using digital technology provides more abundant information to individuals, unlike festivals that lack personalized explanations.

In addition, digital hyperlinks enable an expanded understanding of cultural heritage [40]. Using these advantages of digital, various cultural heritage education using YouTube is being conducted [41], but this has limitations in that it is not a direct participant. Therefore, museums use VR or AR to experience cultural heritage from a first-person perspective, such as the Gangju Chilseok Gosaum Nori Video Experience Center. The virtual tools to exhibit cultural heritage have contributed not just to the enjoyment of the users but also to the actual understanding of its features and dimensions. Furthermore, after having tried the game and the VR experience, the large majority of the participants has experienced a relevant increase both in their interest in cultural heritage and in the recognition of the importance of conserving it [42]. Experiences of digital technologies such as VR and AR have a positive effect on the experiences of not only the younger generation but also the elderly [43,44].

Second, the fact that frequency of visits to local festivals or museums had a positive effect on frequency of visits to the Gwangju Chilseok Gossaum Nori Video Experience Center shows that cultural heritage conservation programs have a mutual effect on each other, providing a new perspective on the role of the museum. Existing research on museum visitors was a survey on differences in visiting age, preferences of visiting museums, and frequency of visits [45–47]. In addition, a research was conducted to examine the motivation and importance of visiting museums for tourism [48,49] as well as the influence of it on the people indirectly experience an era that they had never experienced at museums and how it changed the perception of the time [50]. Other researches have studied the development of network applications for user convenience [51] and the necessity of integrating museums for cultural heritage preservation systems [52]. However, this study shows that raising citizens' interest in the museum itself is important not only to increase the utilization rate of all museums, but also to enhance the overall understanding of cultural heritage.

Third, the purpose of Gwangju citizens' visit to the museum was different depending on their age and their children. McManus [53] said that the purpose of the museum's visit was for communication in the social group. But Dalaney [54] suggests that the museum, she suggests, offers the experience of "Infotainment", with the "History Hall" designed to function as a space for the leisurely consumption of Canadian history and culture. The results of this survey are that, as in Dalaney's study, visits to museums are based on personal needs rather than on the meaning of social participation. Koreans' needs for education were linked to their own education at an age free from child education, for the purpose of children at the age of having children. In addition, leisure was the main purpose of visiting the museum for the age group who were free from children's education. Sevilha Gosling et al. state that museums, with regard to their function of education and cultural provision, provide experiences for the population, and these can result in information, learning and the transformation of visitors [55].

These results showed that the reason for visiting the museum was not only due to understanding cultural heritage but also to various needs of individuals. Therefore, it is important for museums to understand the needs of each age and develop various programs that can be visited by all ages.

## 6. Conclusions

Regarding the conservation of intangible cultural heritage, apprentice-centered preservation, including the spread and exhibition of the heritage through the general public, is certainly important. In particular, intangible cultural heritage with regional characteristics has a higher potential for conservation when citizens' understanding and participation are supported.

Based on the survey analysis, this study examined Gwangju citizens' awareness regarding Gwangju Chilseok Gossaum Nori, and the degree of their participation in festivals and the Video Experience Center as a method of exhibiting the folk game. The results are as follows: First, direct experiences, such as participation in festivals and visits to the Video Experience Center, were important factors in determining citizens' level of awareness. Encouraging the public to participate in festivals and providing opportunities to join in numerous activities for Gwangju Chilseok Gossaum Nori through the Video Experience Center can effectively raise the public's awareness. Second, when analyzing the factors affecting the intention to visit the Video Experience Center, the increase in the participation of Gwangju citizens in various museums and festivals located in the Gwangju area positively affected the visit to the Gwangju Chilseok Gossaum Nori Video Experience Center. In addition, the positive perception of the Center also became an important variable in the intention to visit. Third, the frequency of visits to museums influenced visits to the Video Experience Center, and the purpose of visiting museums varied by age. Through this, it was confirmed that the development of customized programs for various groups is an important factor in enhancing the understanding of intangible heritage.

Preservation of intangible cultural heritage is difficult due to the uncertainty of the continuity of apprentices and weakening of public awareness due to social changes. Various measures and approaches—such as exhibiting intangible cultural heritage and strengthening public awareness—are critical to overcome these difficulties. COVID-19 has presented a unique challenging environment for exhibition. However, the radical development of science and technology referred to as the 4th Industrial Revolution has led to the discovery of other possibilities of exhibition. In such a social climate, to raise the general public's awareness and to find a new and effective method for exhibiting the original form of cultural heritage, it is necessary to understand not only the technology but also the needs of the general public. Therefore, through this study, an attempt was made to examine the case of Gwangju Chilseok Gossaum Nori, as an example of intangible cultural heritage. However, more attempts are required to consider the current status and effects of the exhibition of intangible cultural heritage in South Korea utilizing a broader viewpoint, and the public's interest and awareness of them. This research may be meaningful as fundamental data for future studies.

**Author Contributions:** Conceptualization, M.H.; Funding acquisition, M.H. and Y.Y.; Methodology, M.H. and Y.Y.; Project administration, Y.Y.; Writing—original draft, M.H. and Y.Y.; Writing—review & editing, Y.Y. All authors have read and agreed to the published version of the manuscript.

**Funding:** This work was supported by the Ministry of Education of the Republic of Korea and the National Research Foundation of Korea (NRF-2017S1A6A3A01078538).

**Ethics statement:** All subjects gave their informed consent for inclusion before they participated in the study.

**Data Availability Statement:** Data available on request.

**Conflicts of Interest:** The authors declare no conflict of interest.

## Notes

1. Lee In-hye [22] (p. 72) explains that in developing museum educational programs, local folk and cultural resources are used as primary references. This is because museum users are local people who have a strong desire to learn about the region. This approach explains that in measuring the level of understanding of the local culture, the local people's level of understanding is the most important yardstick.

2. The Cronbach's alpha is a measure for internal consistency, that is, how closely related a set of items are as a group. It is considered to be a measure of scale reliability. The reliability coefficient of 0.70 or higher is considered "acceptable" in most social science analyses.

3. The Likert scale was used in this analysis. Likert scales are rated by respondents on a scale of 5–7. In this questionnaire, all questions were evaluated on a 5-point scale.

4. A *t*-test is a type of statistical test that is used to compare the means of two different populations. To this end, it examines the similarities between populations by comparing the sample means of the independently extracted samples representing the two populations.

5. One-way ANOVA is an analysis method that compares the means of three or more groups. To perform this analysis, there must be a dependent variable that is a continuous variable and one independent variable with three or more categories.

6. The definition of and the guide for playing Gossaum Nori are explained in the second semester of the fifth grade social studies textbook (page 158) in Korea.

7. Yeo·Choi [56] (p. 18) stated that interpretive media influence the motivation to visit museums because they provide useful information.

8. Correlation analysis is a statistical method used to evaluate the degree of relationship between two quantitative variables. A high correlation indicates that two or more variables have a significant relationship with each other, whereas a weak correlation indicates an insignificant relation between the variables.

9. The correlation coefficient expressing the degree of correlation is denoted by "r". When one variable increases, the correlation coefficient reveals whether another variable increases, decreases, or does not change; estimates the degree of the change; and indicates the degree and direction of the relationship between variables. As for correlation analysis criteria, generally, the closer the value is to 1, the stronger the correlation. Values closer to 0.5 indicate normal correlation, and values less than 0.3 indicate weak correlation.

10. This is a method for deriving the optimal regression equation by appropriately combining the addition and removal of independent variables. When evaluating variables individually, this method tests the significance of each variable: If significant, variables are included in the model; otherwise, they are excluded.

11. As a statistical method for analyzing two or more qualitative variables, including categorical variables, the frequency and ratio of the other variables within the range of one qualitative variable are written as a cross-tabulation. It is used to analyze whether there are differences between groups in the population.

12. In a similar fashion, in the case of the Leeum Kids Education Program, art education and museum visit education are organized jointly such that children become familiar with museum visits through the educational program. This is not only a program to nurture potential museum visitors, but also a form of linking museum visits with opportunities for the viewing of exhibitions through educational programs (http://www.leeum.org/html/education/lecture_view.asp; website was visited on 12 August 2021).

13. Our survey revealed a positive correlation between playing online games (2D, 3D, VR, etc.) and believing that the Video Experience Center is helpful in understanding Gwangju Chilseok Gossaum Nori (see Table 5). This finding suggests the importance of various types of museums that reflect changing generational characteristics.

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
