# Peer review of "Digital Exhibition of Intangible Heritage and the Role of Museums in COVID-19 Era—Focusing on Gwangju Chilseok Gossaum Nori Video Experience Center in South Korea"

_heritage, doi:10.3390/heritage5030140_

Round 1
Reviewer 1 Report
This is a good paper and has a compelling contribution to knowledge. However, it must be improved as it currently lack essential and important aspects of the research components, as detailed below:
1. Engagement with literature and latest state of research in this field needs more work with clear referencing of up-to-date literature.
2. There is a need for more theoretical discussion to establish the research field and enuiry.
3. The authors need to give more details on the virtual exhibition and show visuals of how the intangible heritage has been utilised, displayed and engaged with. Currently, The paper does not demonstrate what the experiences are or showcase its components. This limits our understanding of the context of the fieldwork and hence hard to justify the results and findings.
4. The survey questions are very simple and direct. They could have been more elaborated to extract more genuine information.
5. The authors need to contextualise the research findings and conclusions within the field and what new and novel knowledge those results provide.
Author Response
- I presented the revised contents in a table as follows.
|
Comment |
Corrections |
|
Engagement with literature and latest state of research in this field needs more work with clear referencing of up-to-date literature. |
I presented more discussions on the field of research and added references to it. |
|
There is a need for more theoretical discussion to establish the research field and enuiry |
|
|
The authors need to give more details on the virtual exhibition and show visuals of how the intangible heritage has been utilised, displayed and engaged with. Currently, The paper does not demonstrate what the experiences are or showcase its components. This limits our understanding of the context of the fieldwork and hence hard to justify the results and findings. |
Visual information about the virtual exhibition could not be attached due to copyright issues in the museum. However, for a better understanding of Gwangju Chilseok Gos-saum Nori, a photo was attached. <Figure 1> |
|
The survey questions are very simple and direct. They could have been more elaborated to extract more genuine information. |
I agree with you. However, this study was designed for regression analysis related to citizen perception of Gwangju Chilseok Gos-saum Nori. In addition, as suggested in page 2, various items were investigated in addition to the items presented in Table 2, but only the most relevant representative questions were presented in Table 2. In the future, I will conduct research using a more precise survey on how to preserve intangible heritage. |
|
The authors need to contextualise the research findings and conclusions within the field and what new and novel knowledge those results provide. |
In order to analyze the meaning of the results of this study based on existing studies, a Discussion chapter was newly organized. |
Reviewer 2 Report
This manuscript is a very rigorous study which will largely contribute to the development of the preservation of cultural intangible heritage according the methods and novelties of the 21st century. This is a way to preserve intangible heritage and pass it to the younger generations.
In my opinion the manuscript should be published as it is.
Author Response
Thank you for your encouraging comments. Here I am uploading the revised manuscript based on the comments of other reviewers.
Reviewer 3 Report
This is an interesting study and the role of museums in COVID-19 period is an important dimension in current tourism studies. However, there are a few items on which the authors have to develop during the revision
1) The abstract should include the word COVID-19, because this stands in the title of the paper. Also, the name of the country, South Korea should appear in the abstract.
2) The introduction should better highlight how this article brings new angles of research in tourism studies or how it pushes forward what we already know in museum tourism.
Moreover, the current version of the introduction of this paper has quite few mentions of museum studies and the Korean case presented in this paper has to better positioned in international museum studies. Even if the international museum studies is large, there are some studies which should be mentioned from previous tourism studies on museums. For instance, it can be shortly presented that museums could be important sites in relation to politics of memory (see Violi, P. Trauma Site Museums and Politics of Memory: Tuol Sleng, Villa Grimaldi and the Boogna Ustica Museum. Theory, Culture, Society, 2012). Then, psychological distance could be important in heritage experinces (see Massara and Severino. Psychological Distance in the Heritage Experience. Annals of Tourism Research, 2013). Furthermore, studies on memorial museums are emerging recently (see Sodaro, A. Exhibiting Atrocity: Memorial Museums and the Politics of Past Violence; Rutgers University Press: New Brunswick, and also Williams, P. Memorial Museums: The Global Rush to Commemorate Atrocities; Berg: Oxford). Finally, it can be mentioned that museums are important for the younger people .
3) The methods and results are nicely presented. Some limitations of the data and methods are needed to be presented.
4) There are no discussions in this paper. I would recommend to be written 3-4 paragraphs of discussions, which should link the findings of this paper to the broader literature review on museum studies.
5) I think that before the conclusions it can be added one paragraph including some policy recommendations and mention that broader urban and rural tourism is changing in Asia, Latin America and North America (see the example for urban Havana in a study of A. Colantonio and R. Potter, 2017 etc), in Western, Central and Eastern Europe (see Light D. el al, 2020 in Journal of Balkan and Near Eastern Studies) etc and new policy solutions during COVID-19 for museums in Koreea and in other countries worldwide can be stipulated for the shorter and longer term.
6) Conclusions should be a bit expanded by mentioning the international implications of this study as well as how other scholars can complement the outcomes of this study.
5) The reference list is short - there are only 32 references at the moment. It would be good the engagement with international literature in museum studies to be enlarged to about 50 references. I proposed some works to be added at my above point 1, but there could be added even more museum-related studies both in the literature review and in the discusssions of this paper.
Author Response
1) The abstract should include the word COVID-19, because this stands in the title of the paper. Also, the name of the country, South Korea should appear in the abstract.
: The abstract has been modified.
2) The introduction should better highlight how this article brings new angles of research in tourism studies or how it pushes forward what we already know in museum tourism.
Moreover, the current version of the introduction of this paper has quite few mentions of museum studies and the Korean case presented in this paper has to better positioned in international museum studies. Even if the international museum studies is large, there are some studies which should be mentioned from previous tourism studies on museums. For instance, it can be shortly presented that museums could be important sites in relation to politics of memory (see Violi, P. Trauma Site Museums and Politics of Memory: Tuol Sleng, Villa Grimaldi and the Boogna Ustica Museum. Theory, Culture, Society, 2012). Then, psychological distance could be important in heritage experinces (see Massara and Severino. Psychological Distance in the Heritage Experience. Annals of Tourism Research, 2013). Furthermore, studies on memorial museums are emerging recently (see Sodaro, A. Exhibiting Atrocity: Memorial Museums and the Politics of Past Violence; Rutgers University Press: New Brunswick, and also Williams, P. Memorial Museums: The Global Rush to Commemorate Atrocities; Berg: Oxford). Finally, it can be mentioned that museums are important for the younger people .
: In Chapter 2, we presented new information regarding the meaning of the Gwangju Chilseok Gosaum Nori video experience center as a museum and the meaning of a new methodology for preserving intangible heritage.
3) The methods and results are nicely presented. Some limitations of the data and methods are needed to be presented.
4) There are no discussions in this paper. I would recommend to be written 3-4 paragraphs of discussions, which should link the findings of this paper to the broader literature review on museum studies.
5) I think that before the conclusions it can be added one paragraph including some policy recommendations and mention that broader urban and rural tourism is changing in Asia, Latin America and North America (see the example for urban Havana in a study of A. Colantonio and R. Potter, 2017 etc), in Western, Central and Eastern Europe (see Light D. el al, 2020 in Journal of Balkan and Near Eastern Studies) etc and new policy solutions during COVID-19 for museums in Koreea and in other countries worldwide can be stipulated for the shorter and longer term.
6) Conclusions should be a bit expanded by mentioning the international implications of this study as well as how other scholars can complement the outcomes of this study.
: Limitations on the research results and discussions related to existing studies are presented in the newly constructed Discussion chapter.
7) The reference list is short - there are only 32 references at the moment. It would be good the engagement with international literature in museum studies to be enlarged to about 50 references. I proposed some works to be added at my above point 1, but there could be added even more museum-related studies both in the literature review and in the discusssions of this paper.
: I presented more discussions on the field of research and added references to it.
Round 2
Reviewer 3 Report
Authors have improved the quality of this paper by adding the discussion section. A minor revision is further required because paper’s literature review and reference list are still quite short.
In my previous review I proposed several studies to be connected to this Korean case, but none of them has been taken into consideration in the current version of the paper. For instance, at page 4, lines 531-532, after sentence ‘Existing research on museum visitors was a survey on differences in visiting age…’ it can be mentioned several more works in museum studies, see the studies on museums of Violi, Sodaro, and several studies of Light D. et al. on assessing the impact of the Sighet memorial museum on young people (eg doi: 10.3390/soc11020043) and the importance of museums for tourism. Also, implications of this study on international museum studies have to be shortly presented in the conclusion section.
I also think that the title of the paper could be ‘Digital Exhibition of Intangible Heritage and the Role of Museums in COVID19 era : The Gwangju Chilseok Gossaum Nori Video Experience Center in South Korea’. So it is important for the readers to know on which Korean state the museum is – South Korea or North Korea (The Democratic People's Republic of Korea).
Author Response
Dear Reviewer
We are pleased to submit the revision of our manuscript entitled “Digital Exhibition of Intangible Heritage and the Role of Mu-seums in COVID19 era - Focusing on Gwangju Chilseok Gos-saum Nori Video Experience Center in South Korea” for your further consideration. I am very grateful that you read our manuscript carefully. We tried to faithfully reflect your constructive comments.
Responses to the Comments of Reviewer
Major points
|
Comment 1. 1) Authors have improved the quality of this paper by adding the discussion section. A minor revision is further required because paper’s literature review and reference list are still quite short. 2) In my previous review I proposed several studies to be connected to this Korean case, but none of them has been taken into consideration in the current version of the paper. For instance, at page 4, lines 531-532, after sentence ‘Existing research on museum visitors was a survey on differences in visiting age…’ it can be mentioned several more works in museum studies, see the studies on museums of Violi, Sodaro, and several studies of Light D. et al. on assessing the impact of the Sighet memorial museum on young people (eg doi: 10.3390/soc11020043) and the importance of museums for tourism. Also, implications of this study on international museum studies have to be shortly presented in the conclusion section. 3) I also think that the title of the paper could be ‘Digital Exhibition of Intangible Heritage and the Role of Museums in COVID19 era : The Gwangju Chilseok Gossaum Nori Video Experience Center in South Korea’. So it is important for the readers to know on which Korean state the museum is – South Korea or North Korea (The Democratic People's Republic of Korea).
|
Response
- Authors have improved the quality of this paper by adding the discussion section. A minor revision is further required because paper’s literature review and reference list are still quite short.
: We added some more references while revising the discus chapter.
- In my previous review I proposed several studies to be connected to this Korean case, but none of them has been taken into consideration in the current version of the paper. For instance, at page 4, lines 531-532, after sentence ‘Existing research on museum visitors was a survey on differences in visiting age…’ it can be mentioned several more works in museum studies, see the studies on museums of Violi, Sodaro, and several studies of Light D. et al. on assessing the impact of the Sighet memorial museum on young people (eg doi: 10.3390/soc11020043) and the importance of museums for tourism. Also, implications of this study on international museum studies have to be shortly presented in the conclusion section.
: Among the two references you provided in the previous review, the book "Urban Tourism and Development in the Socialist State" is not currently on sale in Korea. So unfortunately, we couldn't refer to it because we couldn't read it.
However, in this revision, we were able to discuss further by referring to Light, D., Crete,an, R., & Dunca, A. M. (2021). Thank you very much. You can check this on page 10.
- I also think that the title of the paper could be ‘Digital Exhibition of Intangible Heritage and the Role of Museums in COVID19 era : The Gwangju Chilseok Gossaum Nori Video Experience Center in South Korea’. So it is important for the readers to know on which Korean state the museum is – South Korea or North Korea (The Democratic People's Republic of Korea).
: We revised Korea to South Korea throughout the paper.